



# Dynamic induction control for mitigation of wake-induced power losses: a wind tunnel study under different inflow conditions

Manuel Alejandro Zúñiga Inestroza[1,2], Paul Hulsman[1,2], Vlaho Petrović[1,2], and Martin Kühn[1,2]

[1]Carl von Ossietzky Universität Oldenburg, School of Mathematics and Science, Institute of Physics, 26129 Oldenburg, Germany
[2]ForWind - Center for Wind Energy Research, Küpkersweg 70, 26129 Oldenburg, Germany

**Correspondence:** Manuel Alejandro Zúñiga Inestroza (manuel.zuniga@uni-oldenburg.de)

**Abstract.** Dynamic induction control (DIC), also known as the pulse method, is a wake mixing strategy that has shown promising results for mitigating wake-induced power losses in wind farms. It relies on dynamic collective blade pitching to enhance turbulent mixing, thereby accelerating the wake recovery. Experimental validation of this concept has been primarily limited to single-turbine cases under idealised conditions without shear and negligible turbulence. This paper presents a wind

tunnel study to investigate the wake recovery improvement induced by DIC in single- and two-turbine configurations, as well as potential power gains in a three-turbine array. The study includes experiments under baseline uniform inflow and two realistic atmospheric boundary layer inflows. Short-range continuous-wave lidar measurements are used to remotely map the time-averaged wake characteristics of each turbine in vertical cross-sections at various downstream positions. First, the wake recovery of the upstream turbine is analysed as a function of pitch amplitude and frequency, with the latter expressed by the

dimensionless Strouhal number. Next, the cascading effect of upstream turbine actuation on the wake of a downstream turbine in greedy mode is examined. Finally, wind farm power gains are assessed in a three-turbine setup incorporating a virtual turbine. Compared to the baseline greedy case, improved wake recovery is observed at both the upstream and downstream turbines, solely through upstream turbine actuation across all cases. This improvement is attributed to intensified turbulent mixing driven by DIC, which induces periodic thrust oscillations at both the actively controlled upstream turbine and the

passive downstream turbine. The effect is particularly pronounced at higher pitch amplitude, while differences across Strouhal number remain minor, suggesting stronger control authority through increased pitch amplitude. Despite a decrease in DIC-added wake recovery with increasing inflow turbulence, potential power gains for the wind farm persist. Overall, this study demonstrates consistent benefits and adaptability of DIC under realistic inflow conditions, highlighting its greater potential in low-turbulence environments.

## 1   Introduction

Wake-induced power losses and fatigue loads pose a persistent challenge for wind turbines in wind farms, with power losses exceeding $40\%$ under full-wake overlap conditions (Barthelmie and Jensen, 2010). This arises from the energy extraction process, which exposes downstream turbines to a wind field characterised by reduced momentum and increased turbulence (Porté-Agel et al., 2020). Although technical and economical constraints make wake effects on the whole unavoidable, no



control strategy is typically implemented to mitigate wake interactions between individual turbines, thereby degrading overall
wind farm efficiency (Boersma et al., 2017). To address this, advanced wind farm flow control strategies are being developed
to minimise wake losses and ultimately reduce the levelised cost of energy (Meyers et al., 2022). These strategies often involve
turbine-level actuation (e.g. yaw, pitch) based on optimal control setpoints, which improve the inflow conditions at downstream
turbines (Houck, 2022). Consequently, overall plant performance can benefit through increased power production or reduced
fatigue loads.

Typically, wind farm flow control implementation focuses foremost on wind farm power maximisation (van Wingerden
et al., 2020). To this end, research has primarily centred on wake steering and static induction control strategies (Kheirabadi and
Nagamune, 2019). Wake steering involves intentionally yawing upstream turbines to deflect their wakes away from downstream
turbines, thereby increasing the total power output (Fleming et al., 2015; Hulsman et al., 2024). This concept has received
the most attention and development, but challenges remain due to the strong dependence of optimal setpoints on temporal
variations in wind direction (Dallas et al., 2024) and power loss behaviour (Howland et al., 2020; Hulsman et al., 2022a).
On the other hand, static induction control involves derating upstream turbines through pitch control, torque control, or a
combination of both to reduce their wake deficit and increase the energy available to downstream turbines (Houck, 2022).
Inconsistent simulation results have undermined interest in this concept. However, power gains may still be achievable under
partial wake conditions or with closely spaced turbines (van der Hoek et al., 2019; Zúñiga Inestroza et al., 2024).

Wake mixing strategies based on individual or collective pitching have demonstrated potential for increased wind farm
power production (Meyers et al., 2022). Unlike wake steering and static induction control, these techniques aim to actively
excite wake instabilities, promoting enhanced turbulent mixing and accelerating wake recovery (Houck, 2022). The collective-
pitch-based approach, commonly termed dynamic induction control (DIC) or the pulse method, is the focus of this study and
its literature review. First results from large-eddy simulations (LES) report power gains of up to $16\,\%$ (Goit and Meyers, 2015)
and $7\,\%$ (Goit et al., 2016) in a $10 \times 5$ wind farm with and without entrance effects, respectively. However, these studies used
a computationally expensive optimisation approach that yielded impractical control signals. Building on this, Munters and
Meyers (2018) introduced a simplified method based on sinusoidal thrust variations to mimic the periodic shedding of vortex
rings observed previously by Goit and Meyers (2015). This approach provided power gains of up to $5\,\%$ in a $4 \times 4$ wind farm,
decreasing to approximately $2\,\%$ at higher inflow turbulence. Yılmaz and Meyers (2018) further advanced this by synthesising a
periodic signal based on optimal generator torque and pitch control, achieving power gains of up to $25\,\%$ in a two-turbine setup
under uniform inflow in LES. These gains diminished or even disappeared with increasing turbulence intensity and integral
length scale.

Unfortunately, experimental validation of DIC is currently limited to a few wind tunnel studies. Frederik et al. (2020b) report
power gains of up to $4\,\%$ in a three-turbine setup under two different atmospheric boundary layer (ABL) inflows. Combining
wind tunnel experiments with numerical simulations, Wang et al. (2020) indicate maximum power gains of $3.6\,\%$ in a three-
turbine array with $5\,\%$ inflow turbulence intensity. Based on particle image velocimetry measurements, van der Hoek et al.
(2022) demonstrate improved wake recovery due to DIC applied to a single turbine under uniform inflow. Although no power





gains were achieved in a virtual two-turbine configuration, subsequent results with a second physical turbine showed a gain of
$0.6\,\%$ (van der Hoek et al., 2024).

While initial numerical and experimental investigations into DIC show promising results, several knowledge gaps remain
open regarding the performance of DIC parameters under different inflow and operating conditions, including the level of wake
recovery improvement, achievable wind farm power gains, cascading effects on downstream turbines, the physical mechanisms
behind DIC and ABL flow interactions, the impact on fatigue loads, among others. Further validation through wind tunnel ex-
periments is essential to bridge the gap between numerical simulations and currently absent field tests, offering well-controlled
and repeatable conditions. This paper presents a wind tunnel study to investigate the wake recovery improvement and wind
farm power gains induced by DIC under baseline uniform inflow and two realistic ABL inflows. Specifically, the following
aspects are addressed:

1. exploring the wake recovery improvement as a function of pitch amplitude and frequency;

2. examining the cascading effect of upstream turbine actuation on a downstream turbine and its wake;

3. evaluating the potential wind farm power gains in a three-turbine array.

The rest of the paper is organised as follows: Sect. 2 outlines the experimental methodology, Sect. 3 presents the results, Sect. 4
provides an analysis and discussion, and Sect. 5 summarises the key conclusions of the study.

## 2    Methodology

This section outlines the experimental methodology. Section 2.1 introduces the wind tunnel facility and the active grid used to
generate ABL inflows. Section 2.2 describes the model wind turbine, while Sect. 2.3 explains the greedy and DIC control strate-
gies. Section 2.4 details the flow measurement techniques, including a short-range continuous-wave lidar and complementary
hot wire measurements. Finally, Sect. 2.5 provides an overview of the experimental setup and measurement cases.

### 2.1    Wind tunnel facility

The experiments were carried out in the large wind tunnel at ForWind, University of Oldenburg, Germany. It is a Göttingen-
type wind tunnel with a contraction ratio of 4:1, a cross-section of $3\,\mathrm{m} \times 3\,\mathrm{m}$ and a closed test section length of $30\,\mathrm{m}$. It consists
of five $6\,\mathrm{m}$ long movable segments, allowing operation in open, partially open or closed test section configuration. The airflow
is driven by four $110\,\mathrm{kW}$ fans capable of reaching wind speeds of up to $42\,\mathrm{m\,s^{-1}}$ with turbulence intensity levels below $0.2\,\%$
when no active grid is installed. A cooling system is used to maintain a constant temperature during operation, while the roof
of each test section segment is adjusted to achieve a zero pressure gradient. An active grid can be installed at the wind tunnel
nozzle to generate tailored, reproducible turbulent and sheared inflows. The active grid consists of 80 shafts with square flaps,
which can be independently controlled to modify the local blockage of the flow. This is achieved by dynamically varying the
angle of the flaps according to user-defined motion protocols (Kröger et al., 2018; Neuhaus et al., 2021).





## 2.2 Model wind turbine

Two in-house developed MoWiTO 0.6 model wind turbines were used during the measurement campaign. The rotor blades are designed according to the NREL–5 MW reference turbine (Jonkman et al., 2009), with a geometric scaling factor of 1:217, resulting in a rotor diameter of $0.58\,\mathrm{m}$ and a blockage ratio below $3\,\%$ in the wind tunnel. To account for scaling effects, the blades feature an SD7003 low-Reynolds-number airfoil (Schottler et al., 2016). The nacelle houses a direct current (DC) motor (Faulhaber 3863H048CR) acting as a generator and a stepper motor (Faulhaber AM2224-R3-4.8-36) used for collective blade

pitching, both equipped with an encoder. The DC motor's encoder measures the rotational speed (rpm), while the aerodynamic torque is estimated from the voltage drop across a shunt resistor. The stepper motor's encoder measures the collective blade pitch angle. These values are then used to calculate the electrical power generated by the model wind turbine. Additionally, a field-effect transistor acts as a variable resistor within the circuit, enabling control of the generator torque for rotor speed adjustment. Furthermore, the tower base is instrumented with strain gauges (HBM 1-DY43-3/350) in full Wheatstone bridge

configuration to obtain thrust measurements. Calibration is done by hanging weights on the nacelle's rear through a pulley system at the start of the measuring campaign. All control algorithms (e.g. torque control, pitch control) are executed in real-time on a National Instruments cRIO-9066 system, while acquiring the data at $5\,\mathrm{kHz}$.

## 2.3 Control strategies

Two different control modes were implemented during the experiments: (i) baseline greedy control and (ii) DIC.

### 105 2.3.1 Greedy mode

The greedy control mode follows the conventional $K\omega^2$ control law by Bossanyi (2000), which aims to maximise the turbine's power output in the partial load region. This is achieved by setting the generator torque ($Q_\mathrm{G}$) proportional to the square of the rotor speed ($\omega$), as follows:

$$Q_\mathrm{G} = K\omega^2 = \frac{\rho\pi R^5 C_\mathrm{Q}(\lambda^*,\beta^*)}{2\lambda^{*2}}\omega^2, \tag{1}$$

where $K$ is the controller gain, depending on the air density ($\rho$), rotor radius ($R$), torque coefficient ($C_\mathrm{Q}$) and design tip speed ratio ($\lambda^*$). $C_\mathrm{Q}$ is a function of both $\lambda^*$ and the optimal pitch angle ($\beta^*$). The generator torque is adjusted via a proportional-integral-derivative (PID) controlled external voltage, which ensures the rotor speed is maintained at the point of optimal efficiency. MoWiTO 0.6 uses a set-point function $Q_\mathrm{G}(\omega)$ derived from characterisation experiments conducted at various wind speeds and pitch angles at the start of each measurement campaign. For this campaign, the turbines operated optimally at a tip

speed ratio $\lambda = 5.60$, thrust coefficient $C_\mathrm{T} = 0.86$ and power coefficient $C_\mathrm{P} = 0.37$, at an inflow wind speed of $7\,\mathrm{m\,s^{-1}}$.

### 2.3.2 DIC mode

The DIC mode applies a sinusoidal signal to the collective blade pitch controller to induce periodic thrust oscillations, while the generator torque controller remains active to maintain operation near optimal aerodynamic efficiency. The control signal



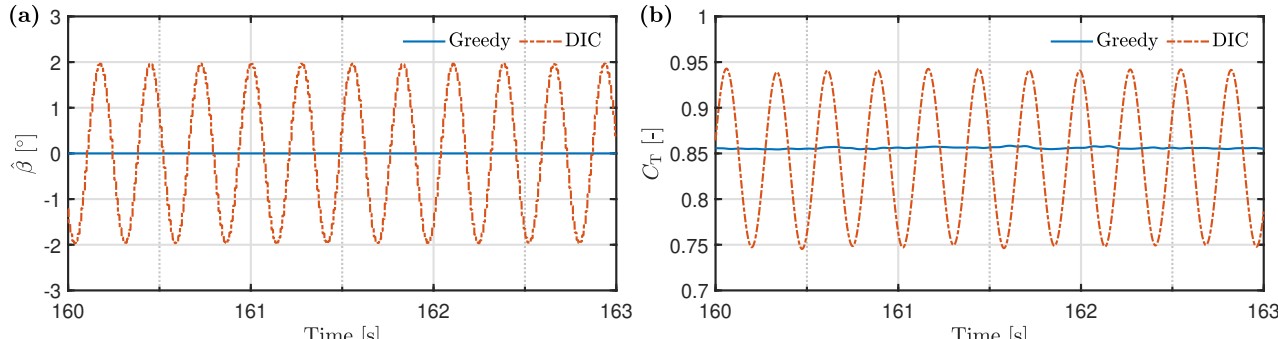

**Figure 1.** Time series excerpt depicting: (a) the collective blade pitch signal and (b) the thrust coefficient under greedy and DIC ($St = 0.30, A = 2°$) operation modes.

to the stepper motor is parameterised by the pitch excitation amplitude ($A$) and frequency ($f_\beta$), the latter typically expressed in terms of the dimensionless Strouhal number ($St$). Here, $St$ represents the ratio of the flow oscillation speed due to periodic pitching to the convective flow speed, defined as:

$$St = \frac{f_\beta D}{u_\infty},\tag{2}$$

where $D$ is the rotor diameter, and $u_\infty$ is the average inflow wind speed at hub height. The dynamic collective blade pitching ($\hat{\beta}$) is then described by:

$$\hat{\beta} = \beta_0 + A\sin\left(2\pi\frac{St\,u_\infty}{D}t\right),\tag{3}$$

where $\beta_0$ is the fine pitch angle that maximises power in the partial-load region, and $t$ is the time. The range of $St$ for this study is chosen based on previous numerical (Munters and Meyers, 2018; Yılmaz and Meyers, 2018) and experimental studies (Frederik et al., 2020b; Wang et al., 2020), which report optimal wake recovery within $St \in [0.24, 0.38]$. Figure 1a illustrates the collective blade pitch signal under both greedy and DIC modes. The DIC signal is implemented at $u_\infty = 7\,\mathrm{m\,s^{-1}}$, with $A = 2°$ and $St = 0.30$ (i.e. $f_\beta = 3.63\,\mathrm{Hz}$). This corresponds to a low-frequency actuation of about $0.017\,\mathrm{Hz}$ at the NREL–5 MW scale, with one cycle completed every $59\,\mathrm{s}$. Figure 1b depicts the $C_\mathrm{T}$ for both control modes, with the DIC case showing a clear harmonic response to the sinusoidal pitch actuation.

## 2.4 Flow measurements

### 2.4.1 WindScanner lidar

Wake measurements were performed with a short-range continuous-wave WindScanner lidar, developed and manufactured by the Technical University of Denmark (DTU). This remote sensing device provides highly spatially resolved flow measurements over large measurement planes in a flexible manner, without flow disturbance. Its effectiveness has been demonstrated in previous wind tunnel experiments combined with model wind turbines (e.g., van Dooren et al., 2017; Hulsman et al., 2020, 2022b;





Zúñiga Inestroza et al., 2024), showing good agreement with hot-wire measurements to capture the main trends of average

streamwise velocity field and dissipation rate of turbulence in wind turbine wakes. It is equipped with a steerable scan head, driven by two prism motors and a focus motor, enabling precise focus point adjustments. This allows for measurements in both staring and scanning modes, following user-defined trajectories. The WindScanner uses a coherent detection method to determine the Doppler frequency shift in the backscattered signal of a focused laser beam within a probe volume. At the start of each measurement campaign, the focus point is calibrated by determining the lidar's location and orientation relative to

the experimental setup using a Leica total station. Additionally, an infrared detector card is used to verify the commanded measurement points. Further details on the use and setup of WindScanners in wind tunnel experiments can be found in (van Dooren et al., 2017; Hulsman et al., 2022b).

In this study, a single WindScanner was used to retrieve the local streamwise velocity component ($u$) of the wind velocity vector at a sampling rate of $451.7\,\mathrm{Hz}$. This approach neglects the lateral ($v$) and vertical ($w$) components, potentially introduc-

ing relative errors of up to $2.4\,\%$ (Hulsman et al., 2022b). Taking this into account, $u$ can be derived according to:

$$u = \frac{1}{\cos\chi\cos\delta}\left(v_{\mathrm{LOS}} - v\sin\chi\cos\delta - w\sin\delta\right) \approx \frac{v_{\mathrm{LOS}}}{\cos\chi\cos\delta},\tag{4}$$

where $v_{\mathrm{LOS}}$ is the measured line-of-sight wind speed along the laser's beam direction, $\chi$ and $\delta$ are the azimuth and elevation angles of the line-of-sight direction, respectively.

During the measurement campaign, the WindScanner was used to obtain time-averaged wake characteristics in vertical

cross-sections at downstream distances $x/D \in \{2,4,5,7,9\}$. The coordinate system is defined at the rotor centre of the first turbine, where $x$, $y$ and $z$ are the longitudinal, transversal, and vertical directions, respectively. Figure 2a provides a schematic of the experimental setup (described in Sect. 2.5), with red dashed lines indicating the wake measurement positions. Each vertical scan followed a Lissajous scanning trajectory comprising of 5000 points (Figure 2b), with a period of $15\,\mathrm{s}$ per scan. The scanning process was repeated for $10\,\mathrm{min}$ per case, resulting in over 40 scans per vertical plane at every downstream

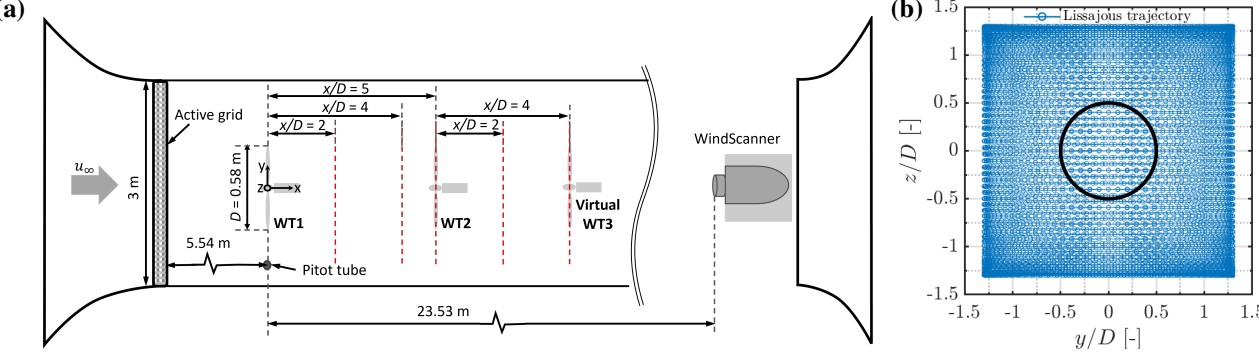

**Figure 2.** (a) Top-view schematic of the experimental setup (not to scale), detailing the arrangement of the model wind turbines and the wake measurement positions (red dahsed lines). (b) Lissajous scanning trajectory used by the WindScanner to map the wake in vertical cross-sections at each downstream location.



distance. Postprocessing of the data included interpolation and averaging onto a grid with $0.13\,D$ by $0.13\,D$ cells, resulting in at least 300 data points per grid cell.

### 2.4.2   Hot-wire array

An important limitation when using WindScanner measurements in a wind tunnel is the probe volume averaging effect, which limits the accuracy of turbulence estimates. Since the laser focuses on a thin cylindrical volume rather than an infinitesimal

point, turbulent structures with a length scale smaller than the probe volume length are partially filtered out (Uluocak et al., 2024). Therefore, complementary hot-wire measurements were conducted for a limited number of cases under uniform inflow, providing insights into the impact of DIC on wake-added turbulence compared to the greedy case. The array consisted of 19 one-dimensional hot wires mounted on an aluminium structure in a horizontal line at hub height. This covered a range of $y/D \in [-1.25, 1.25]$. A traverse system facilitated the measurement of horizontal wake profiles at $x/D \in \{2, 3, 5\}$. Two

Dantec Dynamics 54N80 multi-channel Constant Temperature Anemometry (CTA) systems were used to sample data at $6\,\mathrm{kHz}$ over a period of $120\,\mathrm{s}$. Calibration was done before and after the measurements against a Prandtl tube positioned approximately $2\,\mathrm{m}$ in front of the traversing structure.

### 2.5   Experimental setup and measurement procedure

The experimental setup involved a wind tunnel configuration with a partially open test section, consisting of four connected

segments with a total length of $24\,\mathrm{m}$. The last segment was removed to install the WindScanner on a steel platform near the diffuser, along with a PALAS AGF 10.0 seeding generator to ensure adequate aerosol circulation for laser backscattering. The measurement campaign comprised experiments with single- and two-turbine configurations. For the single-turbine setup, the first turbine (WT1) was positioned $9.55\,D\,(5.54\,\mathrm{m})$ from the nozzle at the tunnel centreline. This configuration allowed to investigate the dependence of wake recovery on pitch amplitude and Strouhal number. Specifically, the time-averaged wake response

of WT1 was examined with the WindScanner under DIC mode for pitch frequencies corresponding to $St \in \{0.25, 0.30, 0.40\}$, each at amplitudes of $A \in \{1°, 2°\}$, and compared to the greedy control case at $x/D \in \{2, 4, 5\}$. For the two-turbine setup, a second turbine (WT2) was installed $5\,D\,(2.9\,\mathrm{m})$ downstream of WT1. The focus here was on examining the cascading effects of WT1 actuation on the wake of WT2. Thus, the wake of WT2 was examined with the WindScanner for the same DIC cases applied to WT1 and compared against the greedy case at $x/D \in \{7, 9\}$, which corresponds to $x'/D \in \{2, 4\}$, with $x'/D = 0$

defined relative to the position of WT2. Note that WT2 remained in greedy operation throughout the experiments. The inflow wind speed was monitored with a Prandtl tube perpendiculat to WT1's rotor plane at hub height and positioned $0.9\,\mathrm{m}$ to the left of the tunnel centreline. Figure 3a provides an upstream view of the two-turbine setup, while Fig. 3b highlights the installation of the WindScanner.

All experiments were conducted in the partial load region at $u_\infty = (7.0 \pm 0.1)\,\mathrm{m\,s^{-1}}$ for three different inflow conditions:

(i) uniform (no shear), (ii) ABL Type-I ($\alpha = 0.11$), and (iii) ABL Type-II ($\alpha = 0.22$), where $\alpha$ represents the best-fit power law exponent accounting for wind shear across the rotor area. The active grid was installed at the nozzle and used in active mode to generate the ABL inflows, while no grid (empty nozzle) was used for the uniform inflow. The WindScanner was used




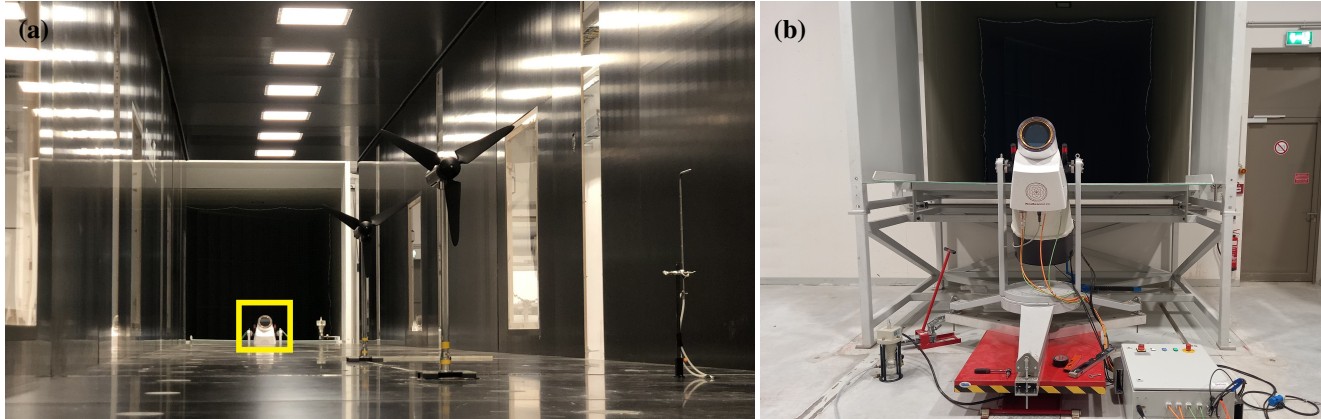

**Figure 3.** (a) Upstream view of the experimental setup with two MoWiTO 0.6 and the WindScanner (yellow square). (b) Close-up of the WindScanner lidar used to remotely map the wake.

to characterise the inflow development along the empty test section at $x/D \in \{0, 2, 5, 7, 9\}$. The vertical profiles of normalised wind speed ($u/u_\infty$) for all three inflow conditions are shown in Fig. 4. Note that $x/D = 9$ is excluded for the uniform inflow

case due to a faulty data set. Each point represent the spatially averaged streamwise velocity component across all horizontal positions ($y/D$) at each height ($z/D$). Despite slight variations for the ABL cases, the wind speed distribution remained fairly stable along the test section for all inflow cases. The inflow turbulence intensity ($TI$) was measured along the tunnel centreline at hub height using the WindScanner's staring mode for both ABL cases, while hot-wire measurements were used for the uniform inflow case. Note that the measured $TI$ using the WindScanner is inherently low-pass filtered due to probe-volume

averaging effects, as explained in Sect. 2.4.2. Table 1 summarises the $TI$ values for all inflows at $x/D \in \{0, 2, 5, 7, 9\}$. The stability of the flow throughout the measurement domain is corroborated under uniform inflow, with an average $TI$ of $0.2\,\%$. For ABL Type-I, $TI$ shows slight downstream variations, while for ABL Type-II, it decays as expected for fully developed

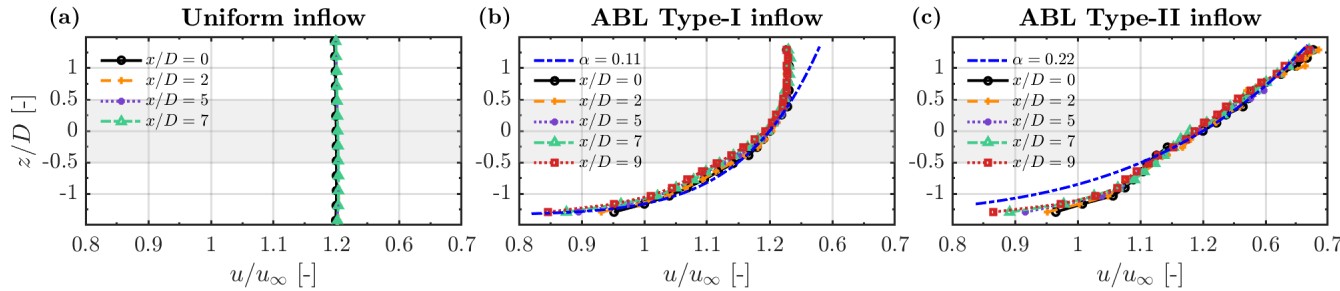

**Figure 4.** Vertical profiles of normalised wind speed ($u/u_\infty$) measured by the WindScanner along the empty wind tunnel test section for different inflow conditions. The grey shaded region represents the rotor area.





**Table 1.** Development of turbulence intensity ($TI$) along the wind tunnel test section at hub height for different inflow conditions, measured using hot wires (HW) or WindScanner staring mode (WS).

| Inflow | $x/D = 0$ | $x/D = 2$ | $x/D = 5$ | $x/D = 7$ | $x/D = 9$ | |
|---|---|---|---|---|---|---|
| Uniform (HW) | 0.2 | 0.2 | 0.2 | 0.2 | 0.2 | [%] |
| ABL Type-I (WS) | 3.8 | 3.9 | 4.0 | 4.0 | 4.1 | [%] |
| ABL Type-II (WS) | 8.4 | 7.8 | 7.5 | 7.2 | 7.0 | [%] |

flows behind a grid. Given the reproducibility of the inflows generated by the active grid, all experiments are expected to have experienced similar $TI$ variations without compromising the validity of the results.

## 3 Results

The experimental results are organised into three main sections. Section 3.1 focuses on the single-turbine setup, analysing the impact of WT1 actuation on wake recovery improvement, wake-added turbulence, and thrust coefficient behaviour. Section 3.2 centres on the two-turbine setup, exploring the cascading effects of WT1 actuation on WT2's wake recovery and thrust coefficient. Section 3.3 evaluates potential wind farm power gains in a three-turbine setup, utilising a virtual turbine in the analysis.

### 3.1 Single-turbine setup: impact of pitch actuation amplitude and Strouhal number

#### 3.1.1 Wake recovery improvement of WT1

The impact of WT1 actuation with DIC on wake recovery is analysed in terms of pitch amplitude and Strouhal number relative to the greedy case across different inflow conditions. Figure 5 illustrates this by presenting wake contours of the normalised streamwise velocity difference between DIC and the baseline greedy case $(u_{\mathrm{DIC}} - u_{\mathrm{Greedy}})/u_\infty$. Each row corresponds to a distinct inflow type, with each subfigure depicting a different downstream position, $x/D \in \{2, 4, 5\}$. Two pitch amplitudes, $A \in \{1°, 2°\}$, are examined under uniform inflow (Fig. 5a-f), while only $A = 2°$ is investigated under both ABL inflows (Fig. 5g-l). Since similar trends are observed for different Strouhal numbers (cf. wind speed deficit profiles of WT1 at hub height in Appendix A1), only cases with $St = 0.30$ are depicted here.

At $x/D = 2$, DIC results in a wind speed deficit (blue regions) at the wake centre and surrounding flow, while increasing the available wind speed in an annulus near the rotor edges (red region). Further downstream, at $x/D \in \{4, 5\}$, this annulus gradually expands to encompass almost the entire swept area of a virtual turbine (indicated by a black circumference), with the magnitude of increased wind speed also rising, especially at the higher amplitude of $A = 2°$. Additionally, the wind speed deficit in the surrounding flow becomes more pronounced, concentrating in a semiannular-shaped region. These patterns are consistent under both uniform and ABL Type-I inflow conditions, although the ABL case exhibits more pronounced wind speed


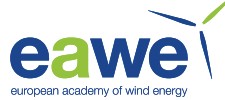


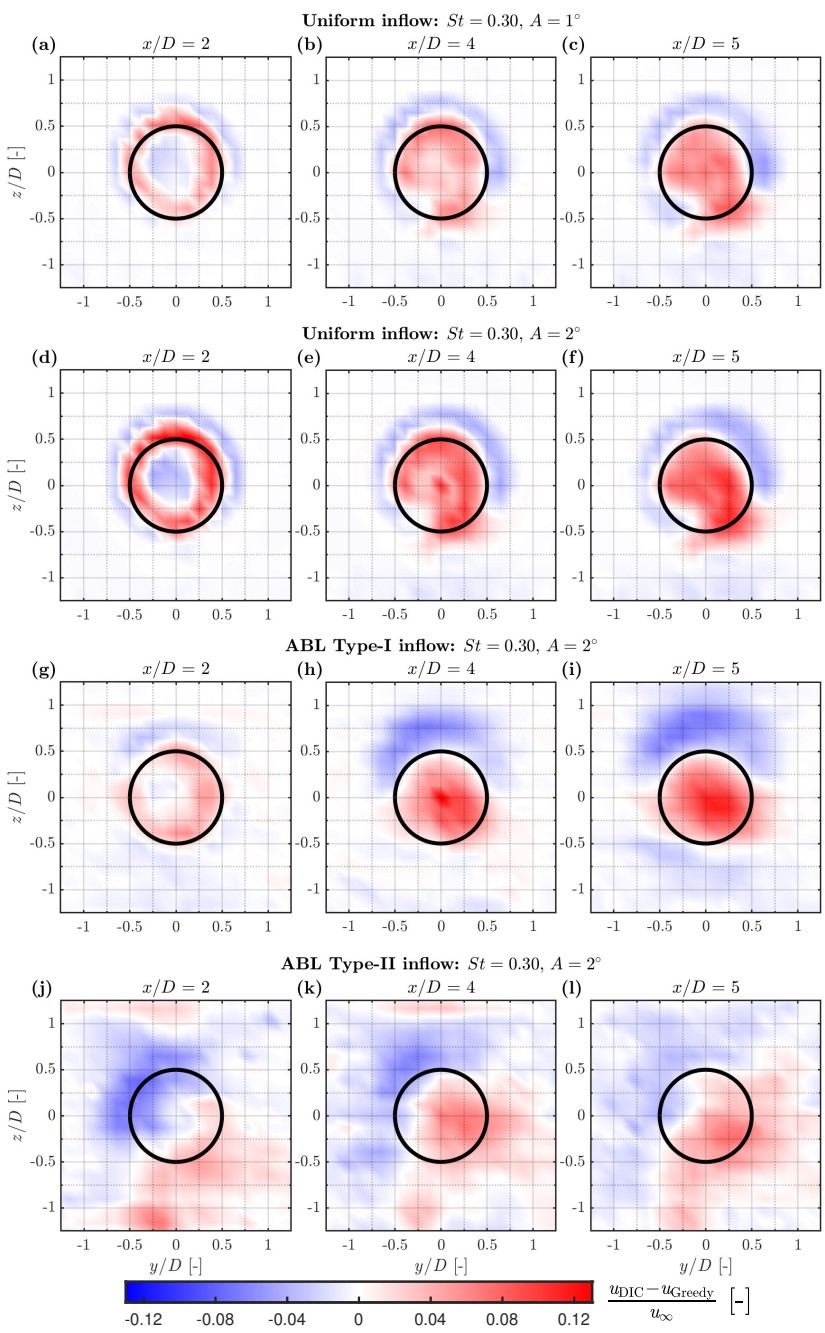

**Figure 5.** Time-averaged wake contours of WT1 depicting the normalised streamwise velocity difference between DIC at $St = 0.30$ and the baseline greedy case $(u_{\mathrm{DIC}} - u_{\mathrm{Greedy}})/u_\infty$. Each row represents a different inflow condition at downstream locations $x/D \in \{2, 4, 5\}$. The black circumference outlines the edges of a virtual downstream turbine. The wake is visualised from the WindScanner perspective, looking upstream towards WT1 with a clockwise wake rotation.





deficit patches, particularly in the upper region of the velocity field, due to the inflow wind shear. For the ABL Type-II inflow, DIC causes a markedly asymmetric distribution of relative changes in the wake velocity field. At $x/D = 2$, the wind speed deficit is concentrated in the second quadrant, while patches of increased wind speed appear in the fourth. The quadrants are defined counterclockwise from the positive $x$-axis, following standard mathematical convention. Further downstream, the wind

speed deficit diffuses more into the surrounding flow compared to previous cases, while regions of higher wind speed spread across the rotor area, remaining predominantly deflected to the right. Potential causes of this asymmetric wake behaviour are discussed in Sect. 4.

To quantify the wake recovery improvement due to WT1 actuation, the normalised rotor equivalent wind speed ($u_{\mathrm{REWS}}/u_\infty$) as a function of downstream distance is computed assuming a full-wake overlap scenario (Fig. 6). Each subchart corresponds

to a different inflow type, with groups of four bars comparing greedy and DIC cases at downstream positions $x/D \in \{2, 4, 5\}$. The weighted average wind speed provides a better estimate of the energy available in the wake, which could be captured by a downstream turbine. For this purpose, the wake within the rotor area is divided into five ring segments based on the measured wake cross-sections, as follows:

$$u_{\mathrm{REWS}} = \sqrt[3]{\sum_{i=1}^{5} u_i^3 \frac{A_i}{A_{\mathrm{R}}}}, \qquad (5)$$

where $u_i$ is the average wind speed within the $i$-th ring segment, $A_i$ is the area of the $i$-th ring segment, and $A_{\mathrm{R}}$ is the rotor swept area. Percentage values indicate the relative change in $u_{\mathrm{REWS}}/u_\infty$ due to DIC compared to the greedy case, with bold text highlighting the optimal recovery. The red error bars indicate the rotor average uncertainty of $u_{\mathrm{REWS}}/u_\infty$, calculated according to the standard uncertainty propagation method used in (van Dooren et al., 2017; Hulsman et al., 2022b).

Experiments under uniform inflow reveal a dependence of wake recovery on pitch amplitude and Strouhal number. For

$A = 1°$, the optimal wake recovery relative to the greedy case occurs at the highest frequency tested ($St = 0.40$) across all downstream positions, reaching a peak of 7.9 % at $x/D = 5$. In comparison, for $A = 2°$, the optimal $St$ varies with downstream distance: $St = 0.40$ at $x/D = 2$, a similar recovery for $St \in \{0.30, 0.40\}$ at $x/D = 4$, and $St = 0.30$ at $x/D = 5$. The highest recovery (10.2 %) is achieved at $x/D = 4$. Noteworthy is that the magnitude of wake recovery improvement is more responsive to increased amplitude than to changes in Strouhal number. For ABL Type-I inflow, the optimal recovery consistently aligns

with $St = 0.30$ at all downstream distances, with a maximum improvement of 8.8 % at $x/D = 5$. For ABL Type-II inflow, the optimal recovery is also achieved with $St = 0.30$, reaching a peak of 3.8 % at $x/D = 4$, while no benefit is attained at $x/D = 2$ due to the asymmetry described earlier.

### 3.1.2 DIC-added turbulence of WT1

To analyse the impact of DIC on added turbulence relative to the baseline greedy case, Fig. 7 presents horizontal wake profiles

of local turbulence intensity ($TI$) at hub height for downstream distances $x/D \in \{2, 3, 5\}$ under uniform inflow conditions. These profiles provide indication of the wind speed fluctuations within the wake, derived from hot-wire measurements as





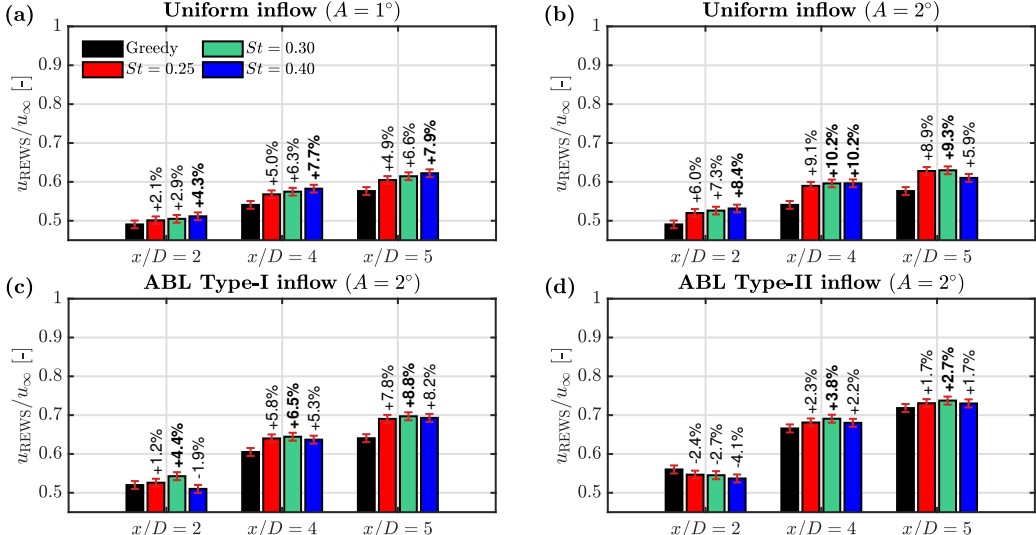

**Figure 6.** Wake recovery of WT1 expressed by the normalised rotor equivalent wind speed ($u_{\mathrm{REWS}}/u_\infty$). Each subchart corresponds to a different inflow type, with groups of four bars comparing greedy and DIC cases at downstream positions $x/D \in \{2, 4, 5\}$. Percentage values indicate the relative change in $u_{\mathrm{REWS}}/u_\infty$ due to DIC compared to the baseline greedy case. Bold text highlights the optimal recovery.

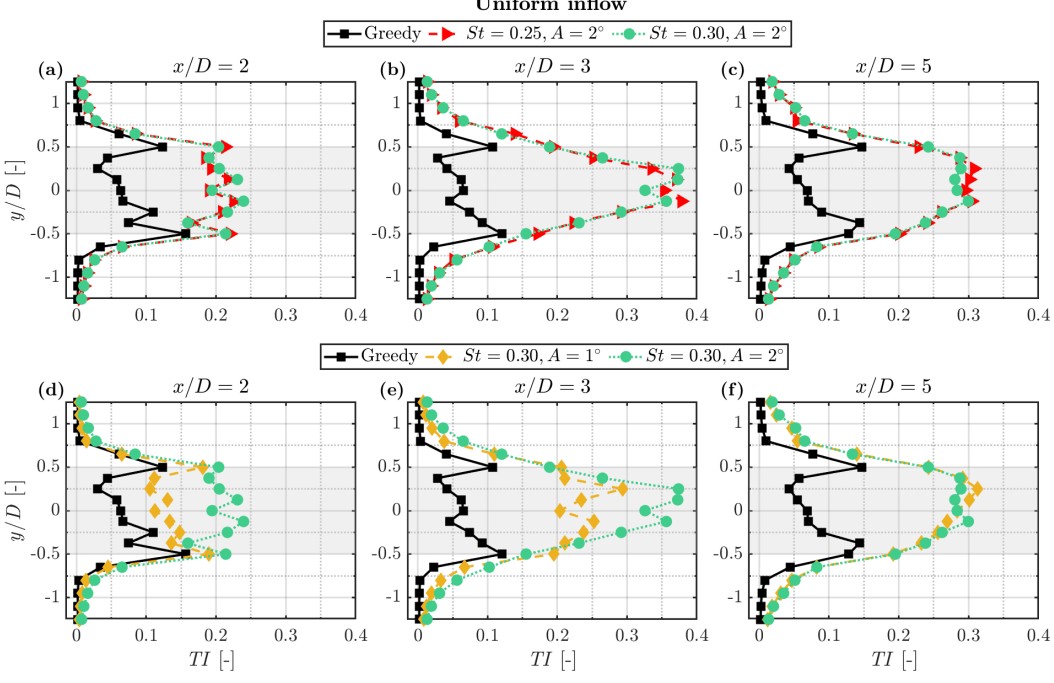

**Figure 7.** Horizontal wake profiles of WT1 showing local turbulence intensity ($TI$) at hub height under uniform inflow. The top row compares the effect of increasing Strouhal number at a fixed amplitude of $A = 2°$ with the greedy case, while the bottom row depicts the impact of increasing amplitude at the same Strouhal number of $St = 0.30$.





follows:

$$TI(y) = \frac{\sigma(y)}{< u(y) >},\tag{6}$$

where $\sigma(y)$ is the local standard deviation and $< u(y) >$ is the local mean wind speed, both as functions of the transversal position $y$. The top row illustrates the effect of increasing Strouhal number $St \in \{0.25, 0.30\}$ at $A = 2°$, while the bottom row shows the effect of increasing amplitude $A \in \{1°, 2°\}$ at $St = 0.30$. Compared to the greedy operation, all DIC cases display a significant increase in local turbulence and an earlier transition to the far wake region. The latter is typically characterised by the merging of the $TI$ profile into a single peak at the wake centre. Note that the differences in local $TI$ among DIC cases with increasing Strouhal numbers at the same amplitude are marginal (Fig. 7a-c). Conversely, the increase in local $TI$ is more pronounced for DIC cases with higher amplitude at the same Strouhal number (Fig. 7d-f). Likewise, the transition to the far wake region occurs more rapidly with $A = 2°$ compared to the case with $A = 1°$, as indicated by the single peak $TI$ profile, along with a local maximum around $x/D = 3$. Further downstream, at $x/D = 5$, $TI$ is already decaying for the DIC cases, whereas for the greedy case the merging has not yet occurred and instead the local $TI$ is still building up.

### 3.1.3 Thrust coefficient of WT1

To qualitatively illustrate the periodic $C_T$ oscillations induced by collective blade pitching, Fig. 8a-c presents time series excerpts under greedy and DIC ($St = 0.30$) operation modes across different inflow conditions. For comparison, the time series are synchronised to analyse the turbine behaviour when exposed to the same varying inflow conditions. Additionally, the signals are low-pass filtered at $12\,\mathrm{Hz}$ to remove high-frequency noise. Here, $C_T$ is calculated according to:

$$C_T = \frac{F_T}{0.5\rho\pi R^2 u_\infty^2},\tag{7}$$

where $F_T$ is the instantaneous thrust force measured via strain gauges at the tower base of WT1, $\rho$ is the average air density, $R$ is the rotor radius, and $u_\infty$ is a $10\,\mathrm{min}$ averaged wind speed measured by the Prandtl tube at hub height. Note that in cases with ABL inflow, this point measurement does not fully represent inflow variations across the rotor area. In general, DIC is effectively translated into periodic periodic oscillations of $C_T$ across all inflow conditions. To quantify these variations, Table 2 summarises the mean $C_T$ of WT1, averaged over $10\,\mathrm{min}$ for each case and inflow condition, with the standard deviation shown in parentheses next to the mean value. Compared to the greedy case, the relative change in mean $C_T$ across different $St$ cases is marginal at $A = 1°$ under uniform inflow, while a slight decrease is observed for DIC cases with $A = 2°$. Notably, the standard deviation becomes more pronounced with increasing amplitude. With negligible $TI$ under uniform inflow, the standard deviation relative to the greedy case increases drastically by approximately $500\,\%$ at $A = 1°$ and $900\,\%$ at $A = 2°$. For experiments under both ABL inflow types, the mean $C_T$ is moderately higher for all DIC cases, being most pronounced at $St = 0.40$. The sheared and turbulent nature of ABL inflows results in higher standard deviation for the baseline greedy case, reducing the relative variation increase for DIC cases to $63\,\%$ for ABL Type-I and $33\,\%$ for ABL Type-II.





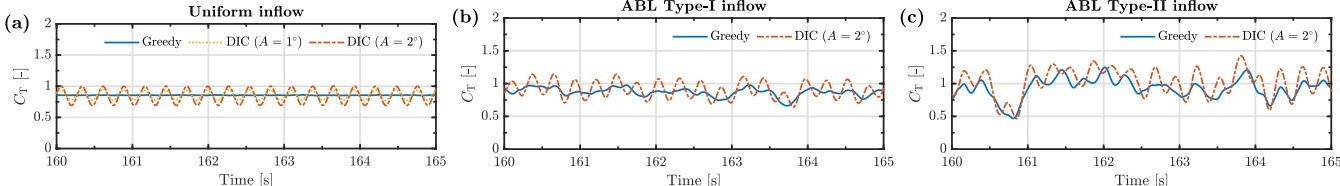

**Figure 8.** Time series excerpts depicting WT1's thrust coefficient ($C_{\mathrm{T}}$) under greedy and DIC ($St = 0.30$) operation modes across different inflow conditions.

**Table 2.** Mean thrust coefficient ($C_{\mathrm{T}}$) of WT1 for greedy and DIC cases across different inflow conditions, with standard deviation in parentheses.

|  | Uniform inflow | | | | ABL Type-I | | ABL Type-II | |
| --- | --- | --- | --- | --- | --- | --- | --- | --- |
| $St$ [-] | $A = 1°$ | | $A = 2°$ | | $A = 2°$ | | $A = 2°$ | |
| Greedy | 0.86 | (0.01) | 0.86 | (0.01) | 0.88 | (0.08) | 0.91 | (0.15) |
| 0.25 | 0.86 | (0.06) | 0.84 | (0.10) | 0.90 | (0.13) | 0.93 | (0.20) |
| 0.30 | 0.86 | (0.06) | 0.84 | (0.10) | 0.92 | (0.13) | 0.93 | (0.20) |
| 0.40 | 0.85 | (0.06) | 0.84 | (0.10) | 0.93 | (0.12) | 0.95 | (0.20) |

## 3.2 Two-turbine setup: cascading effects of upstream turbine actuation on a downstream turbine

### 3.2.1 Wake recovery improvement of WT2

To investigate the cascading effects of WT1 actuation via DIC on WT2's wake recovery, Fig. 9 shows wake contours of the
normalised streamwise velocity difference between cases with and without WT1 actuation $(u_{\mathrm{DIC}} - u_{\mathrm{Greedy}})/u_\infty$. Each row corresponds to a distinct inflow condition, and each subfigure represents specific downstream locations $x'/D \in \{2, 4\}$, where $x'/D = 0$ is defined relative to the position of WT2. Note that WT2 operates in greedy mode across all cases. Since similar trends are observed across different Strouhal numbers (cf. wind speed deficit profiles of WT2 at hub height in Appendix A2), only cases with WT1 actuation at $St = 0.30$ are presented here. In general, WT1 actuation consistently enhances the wake
recovery of WT2 across all inflow conditions. Notably, the wake exhibits regions of increased available wind speed (red regions) spanning almost the entire rotor area of a virtual downstream turbine. This area extends to the right, while surrounded by wind speed deficit patches (blue regions) in the upper and lower regions. These patterns are consistent across all DIC cases, becoming more pronounced at $A = 2°$. Under ABL inflow, the effects remain visible but are more diffused for the ABL Type-II inflow.

To quantify the wake recovery improvement of WT2, Fig. 10 presents bar charts of the normalised rotor equivalent wind speed ($u_{\mathrm{REWS}}/u_\infty$) at downstream distances $x'/D \in \{2, 4\}$. Each subchart corresponds to a distinct inflow type, with groups of four bars comparing cases with and without WT1 actuation. Under uniform inflow, the magnitude of wake recovery improves

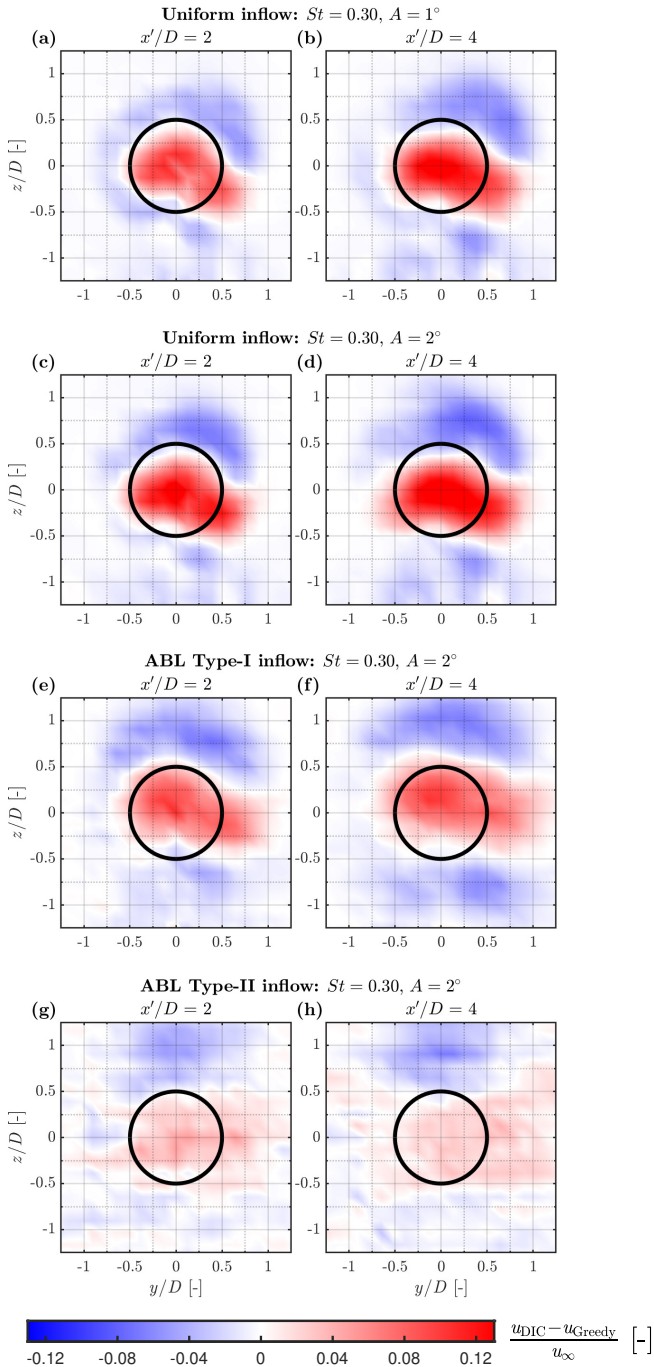

**Figure 9.** Time-averaged wake contours of WT2 depicting the normalised streamwise velocity difference between cases with and without WT1 actuation $(u_{\mathrm{DIC}} - u_{\mathrm{Greedy}})/u_\infty$. Each row corresponds to a different inflow condition at downstream locations $x'/D \in \{2, 4\}$, with $x'/D = 0$ defined relative to the position of WT2. The black circumference outlines the edges of a virtual downstream turbine. The wake is visualised from the WindScanner perspective, looking upstream towards WT2 with a clockwise wake rotation. WT2 remains in greedy operation throughout.





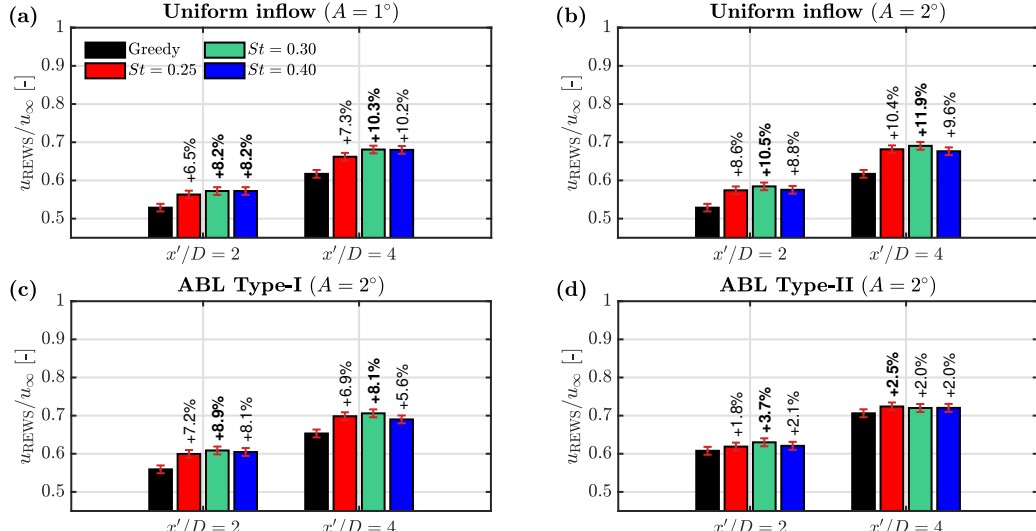

**Figure 10.** Wake recovery of WT2 expressed by the normalised rotor equivalent wind speed ($u_{\mathrm{REWS}}/u_\infty$) at downstream positions $x'/D \in \{2, 4\}$. Each subchart corresponds to a different inflow type, with groups of four bars comparing cases with and without WT1 actuation. Percentage values indicate the relative change in $u_{\mathrm{REWS}}/u_\infty$, with bold text highlighting the optimal recovery.

with increasing amplitude and downstream distance, peaking at $St = 0.30$ and $x'/D = 4$, with a increase of $10.3\%$ at $A = 1°$ and $11.9\%$ at $A = 2°$. A similar trend is observed under ABL Type-I inflow, though with slightly smaller improvement, reaching a maximum of $8.1\%$. For the ABL Type-II inflow, the wake recovery improvement is further reduced, reaching a maximum of $3.7\%$ with $St = 0.30$ at $x'/D = 2$ and $2.5\%$ with $St = 0.25$ at $x'/D = 4$.

### 3.2.2 Thrust coefficient of WT2

The cascading effects of WT1 actuation on WT2 are analysed here in terms of thrust variations and average. Fig. 11 displays $C_{\mathrm{T}}$ time series excerpts of WT2, comparing cases with ($St = 0.30$) and without WT1 actuation across different inflow conditions. Compared to the greedy case, the $C_{\mathrm{T}}$ of WT2 exhibits periodic fluctuations in response to the harmonic actuation of WT1. To quantify these variations, Table 3 summarises the mean $C_{\mathrm{T}}$ of WT2 across all investigated cases, with standard deviations shown in parentheses. Due to the improved wake recovery from WT1 actuation, both the mean $C_{\mathrm{T}}$ and standard deviation of WT2 increase across all conditions. However, the effect is less pronounced under ABL inflow, particularly for the ABL Type-II. Under uniform inflow, the standard deviation increases by up to $233\%$ at $A = 1°$ and $300\%$ at $A = 2°$. Under ABL inflow, the fluctuation increase reduces to $100\%$ for ABL Type-I and $14\%$ for ABL Type-II.

### 3.3 Wind farm power gains

This section evaluates the potential wind farm power gains arising from WT1 actuation with DIC relative to the baseline greedy case. The analysis focuses on a three-turbine array under fully-waked conditions, with a longitudinal spacing of $5\,D$ between



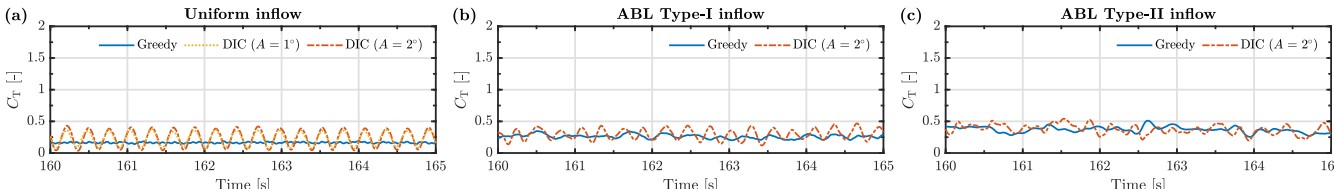

**Figure 11.** Time series excerpts depicting the impact of WT1 under greedy and DIC operation modes on WT2's thrust coefficient ($C_\mathrm{T}$) across different inflow conditions. WT2 remains in greedy operation throughout.

**Table 3.** Mean thrust coefficient ($C_\mathrm{T}$) of WT2 with and without WT1 actuation across different inflow conditions. Standard deviation is shown in parentheses. WT2 operates in greedy mode throughout.

|  | Uniform inflow | | | | ABL Type-I | | ABL Type-II | |
|---|---|---|---|---|---|---|---|---|
| $St$ [-] | $A = 1°$ | | $A = 2°$ | | $A = 2°$ | | $A = 2°$ | |
| Greedy | 0.17 | (0.03) | 0.17 | (0.03) | 0.26 | (0.04) | 0.36 | (0.07) |
| 0.25 | 0.22 | (0.09) | 0.23 | (0.12) | 0.29 | (0.08) | 0.37 | (0.08) |
| 0.30 | 0.23 | (0.10) | 0.23 | (0.12) | 0.30 | (0.08) | 0.38 | (0.08) |
| 0.40 | 0.23 | (0.09) | 0.23 | (0.11) | 0.29 | (0.07) | 0.37 | (0.08) |

WT1 and WT2, and $4\,D$ between WT2 and WT3. The first two turbines are physically installed in the tunnel, while the third
is a virtual turbine based on WindScanner data. The $4\,D$ spacing for WT3 was constrained by the WindScanner's minimum
focus distance. The power of WT3 is calculated based on the previously computed $u_\mathrm{REWS}$, assuming it operates with the same
optimal efficiency as WT1 (i.e. $C_\mathrm{P} = 0.37$) for all cases, without accounting for Reynolds number effects. Note that only WT1
alternates between greedy and DIC modes, while WT2 and WT3 remain in greedy operation throughout. Fig. 12 displays the
mean power ratio ($P_i/P_\mathrm{ref}$) for each turbine (WT1, WT2, WT3) and for the entire wind farm (WF), based on a $10\,\mathrm{min}$ average
per case. Each subfigure corresponds to a different inflow type, with $P_\mathrm{ref}$ representing the total wind farm power output under
greedy operation. Percentage values indicate the relative change in power due to DIC implementation compared to the greedy
case, with bold text highlighting the optimal gains.

Experiments under uniform inflow show that the power loss at WT1 increases with amplitude but remains similar across
Strouhal numbers, peaking at $3.4\,\%$ for $A = 1°$ and $7.3\,\%$ for $A = 2°$. Next, the highest power gains at WT2 occur at $St = 0.40$
for $A = 1°$ and $St = 0.30$ for $A = 2°$, with the optimal gain nearly doubling when moving from the lower to the higher
amplitude. Specifically, it reaches $63.0\,\%$ at $A = 1°$ and $118.2\,\%$ at $A = 2°$. The same behaviour is observed in terms of wake
recovery improvement in Fig. 6. Furthermore, the power gain at WT3 is highest at $St = 0.30$ for both amplitudes, with slightly
higher gains at $A = 2°$, reaching $34.3\,\%$ at $A = 1°$ and $40.1\,\%$ at $A = 2°$. Overall, the combined wind farm power gain reaches
up to $7.7\,\%$ and $9.4\,\%$ for the $A = 1°$ and $A = 2°$ cases, respectively.

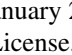 

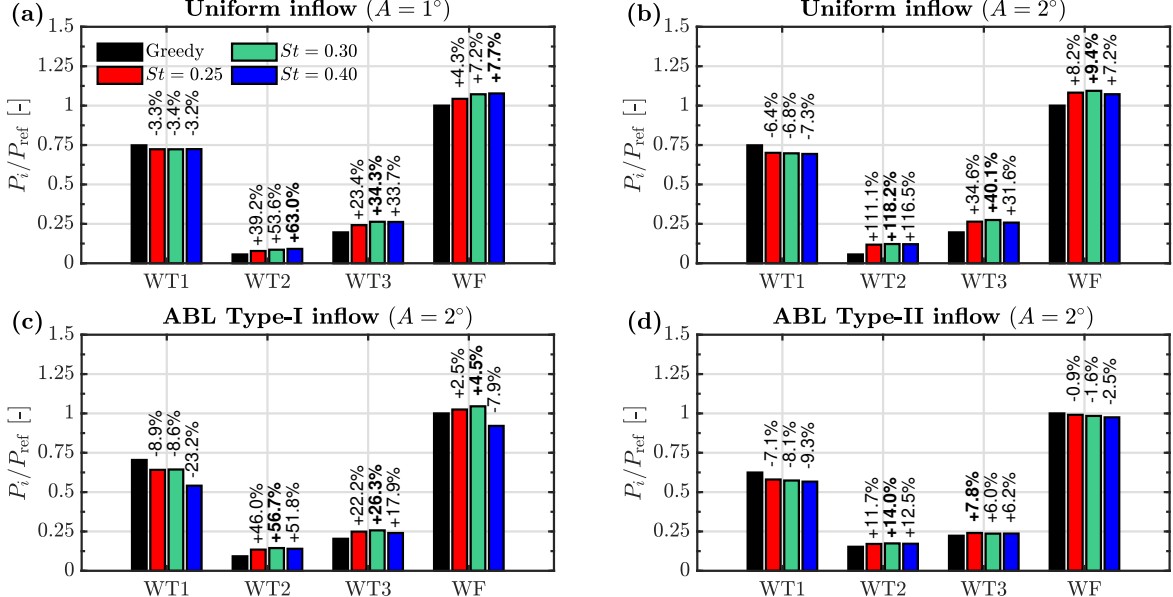

**Figure 12.** Power ratio ($P_i/P_\mathrm{ref}$) of individual turbines (WT1, WT2, WT3) and wind farm (WF) for both greedy and DIC cases, with $P_\mathrm{ref}$ representing the total wind farm power output under greedy operation. Each subchart corresponds to a different inflow type. Percentage values indicate the relative change in $P_i/P_\mathrm{ref}$ due to DIC compared to the greedy case. Bold text highlights the optimal gains.

For the experiments under ABL inflow, the power loss at WT1 remains similar to that observed in uniform inflow cases, except for the DIC case with $St = 0.40$ under ABL Type-I, which exhibits a significantly higher loss of $23.2\,\%$. This may be attributed to unfavorable interactions (e.g., vibrations or resonances) between the turbine actuation at this particular setting and the turbulence characteristics of ABL Type-I. For this inflow, the power gains at WT2, WT3, and WF align with $St = 0.30$, reaching up to $56.7\,\%$, $26.3\,\%$, and $4.5\,\%$ respectively. For the ABL Type-II inflow, WT2 and WT3 achieve smaller power

gains of up to $14.0\,\%$ and $7.8\,\%$, respectively, with no net gain at the farm level.

## 4 Discussion

This study investigated the wake recovery improvement induced by DIC as a function of pitch amplitude and Strouhal number, along with its cascading effects on a downstream turbine operating in greedy mode and the resulting wind farm power gains. To this end, wind tunnel experiments were conducted using single- and two-turbine configurations to map the wake response

of each turbine under baseline uniform inflow and two realisitic ABL conditions.

Compared to conventional wind tunnel techniques such as particle image velocimetry (PIV) and laser Doppler anemometry (LDA), WindScanner measurements enabled flexible and efficient remote mapping of each turbine's wake at multiple downstream locations with high spatial resolution. Depending on the experimental setup and WindScanner operation mode (staring or scanning), phase-averaged characteristics, turbulence dissipation rates, spectral or time series data can be analysed,



albeit constrained by the maximum temporal resolution of $451.7\,\mathrm{Hz}$ and the probe-volume averaging effect (Hulsman et al., 2022b; van Dooren et al., 2022). While this study focused on time-averaged wake characteristics, complementary hot-wire measurements provided indication on the level of DIC-added local turbulence.

In general, the experimental results show that DIC enhances the wake recovery of both the actuated WT1 and the greedy WT2 across all cases compared to the baseline greedy case. This effect is particularly pronounced at higher pitch amplitudes
($A = 2°$), while variations across different Strouhal numbers remain minor. This finding suggests a stronger control authority achieved through increased forcing amplitude than with a higher Strouhal number. This behaviour is attributed to higher amplitudes of $C_\mathrm{T}$ oscillations, which enhance turbulent mixing by introducing higher levels of DIC-added local turbulence, as evidenced in Sect. 3.1.2. Consistent with numerical studies (Munters and Meyers, 2018; Yılmaz and Meyers, 2018), the wake recovery improvement induced by DIC decreases under turbulent and sheared inflow conditions. This effect is exacerbated by
increasing inflow turbulence, which inherently enhances natural wake mixing and results in higher standard deviations of $C_\mathrm{T}$, even without turbine actuation. This is expected to reduce the control authority of DIC, particularly under unstable atmospheric conditions, and should be explored in future studies. Nonetheless, this study demonstrated improved wake recovery even under highly sheared and turbulent inflow, highlighting DIC's potential adaptability to realistic inflow conditions.

A novel contribution of this research is the evaluation of the cascading effects of WT1 actuation on the wake recovery
of a downstream turbine, operating in greedy mode. Notably, WT2 exhibits periodic oscillations of $C_\mathrm{T}$ in response to WT1 actuation, which are assumed to be the triggering factor for the improved WT2's wake recovery. This effect may be attributed to wake synchronisation with the harmonic forcing of DIC at WT1, which causes WT2 to experience higher average wind speeds accompanied by periodic fluctuations (Yılmaz and Meyers, 2018). Consequently, both the mean $C_\mathrm{T}$ and standard deviation of WT2 increase relative to the baseline greedy case. Similar to single-turbine experiments, the cascading effects decrease with
increased inflow turbulence. Future work should aim to better understand and evaluate the cascading effects of DIC.

Unlike experiments under uniform and ABL Type-I inflow, the wake re-energisation induced by DIC at WT1 exhibits an asymmetric behaviour under ABL Type-II inflow (Fig. 5j-l). This can be partially attributed to the influence of wake rotation under sheared inflow, where high-speed momentum is transported downward and low-speed momentum upward. Consequently, the wake of a non-misaligned, greedy turbine tends to be slightly deflected to the right when viewed from downstream (Fleming
et al., 2014; Gebraad et al., 2016). This effect is particularly enhanced under highly sheared and turbulent inflow but appears to be disturbed by DIC. Furthermore, DIC enhances the mechanical turbulence induced by the nacelle and tower (van der Hoek et al., 2022). Since $TI$ levels below hub height are typically reduced relative to undisturbed ABL inflow (Porté-Agel et al., 2020), this additional $TI$ may explain the increase in available wind speed, initially concentrated below hub height and gradually spreading over the rotor area further downstream. Future studies should further explore the complex turbulent
interactions between DIC-actuated turbines and ABL inflows.

Another important aspect to highlight is that, compared to previous studies based on numerical simulations with full-scale turbines (e.g. Yılmaz and Meyers, 2018; Frederik et al., 2020a; Coquelet et al., 2022), the power losses due to DIC in the model wind turbine are more pronounced (approximately $10\,\%$), especially when higher amplitude is combined with a higher Strouhal number. This issue has also been observed in previous wind tunnel studies (e.g. Frederik et al., 2020b; van der





Hoek et al., 2022, 2024), which attribute it to a steeper power curve, typical of model turbines, or to potential vibrations caused by slight rotor imbalances. Nevertheless, it is anticipated that such losses would be smaller in full-scale applications. Supporting this, the aerolastic simulations presented in Appendix B, using the NREL–5 MW reference turbine ($D = 126\,\mathrm{m}$), indicate power losses below $1.6\,\%$ for all DIC cases under both uniform and ABL inflow conditions. Despite the model turbine experiencing higher power losses, the study demonstrated wind farm power gains under both uniform inflow and ABL Type-I

inflow, the latter exhibiting characteristics resembling atmospheric stable conditions. This underscores the potential benefits of DIC implementation in realistic inflow scenarios, particularly for low $TI$ conditions, which typically results in more persistent wakes.

    Interestingly, the optimal wind farm power gains are predominantly achieved with WT1 actuation at $St = 0.30$ and $A = 2°$. This finding closely aligns with the optimal Strouhal number identified in previous wind tunnel studies with three-turbines

(Frederik et al., 2020b; Wang et al., 2020). While the optimal gain at $A = 1°$ was achieved with $St = 0.40$, the net power gains were similar to those obtained with $St = 0.30$. This suggests that adopting a single Strouhal number could simplify integration into a real wind farm controller while providing comparable benefits. Furthermore, the impact of increasing pitch amplitude proved to be more significant, with greater variations in the achievable gains. Although this study was limited to two amplitude cases, it can be inferred that the maximum DIC amplitude should be constrained to limit structural loads, as suggested for the

helix approach (Taschner et al., 2023). In addition, for the adopted setting with full wake overlap and turbine spacings of $5\,D$ and $4\,D$, the results indicate that the largest increase in power occurs at WT2, although considerable gains are also observed at the virtual turbine WT3. Note that the level of power gain at the virtual turbine WT3 is likely underestimated, as the Reynolds number dependency of the turbine's power coefficient was not considered in the analysis.

    Future studies should focus on fatigue load analysis to identify the trade-off between power benefits and structural load

penalties. Frederik and van Wingerden (2022) provide one of the most comprehensive studies examining the impact of wake mixing strategies on different turbine components. Based on FAST and SOWFA simulations, the study shows that DIC primarily affects the tower fore-aft bending moment, with no observed risk of pitch bearing damage compared to conventional individual pitch control (IPC) for load alleviation. Furthermore, a first step before field validation could involve wind tunnel testing with aeroelastic model wind turbines capable of providing meaningful load analysis. Another relevant aspect would

be testing the robustness of the optimal control parameters under dynamic inflow conditions, with varying wind direction, turbulence intensity, and wind speeds.

## 5   Conclusions

This study investigated the potential of DIC to improve wake recovery in single- and two-turbine configurations, as well as potential wind farm power gains in a three-turbine array incorporating a virtual turbine. This encompassed experiments under

baseline uniform inflow and two realistic ABL inflow conditions. WindScanner measurements facilitated the examination of WT1's wake recovery at different pitch amplitudes $A \in \{1°, 2°\}$ and Strouhal numbers $St \in \{0.25, 0.30, 0.40\}$, as well as its cascading effects on the wake of a WT2, operating in greedy mode. Compared to baseline greedy operation, the results show





improved wake recovery of both WT1 and WT2 through sole actuation of WT1. This trend persists across all DIC cases and inflow scenarios, yet the degree of DIC-added recovery decreases with increasing inflow turbulence. The study also provides
insights into the relationship between optimal excitation frequency and pitch amplitude. Notably, stronger control authority is achieved with higher pitch amplitude rather than higher Strouhal number, as increased amplitude enhances turbulent mixing through more pronounced periodic thrust fluctuations. Within the considered range, optimal gains are observed at higher Strouhal numbers for lower pitch amplitudes, and vice versa. This is evidenced by both WindScanner wake measurements and WT2's power output. Furthermore, the study demonstrates wind farm power gains not only under uniform inflow but also
under turbulent and sheared inflow. Overall, this study contributes to validate DIC's potential as an promising wind farm flow control strategy, demonstrating consistent benefits and adaptability under realistic inflow conditions.

Future research should include fatigue load analyses to balance power gains with structural load penalties, as increased pitch parameters, either independently or together, can exacerbate fatigue loading and power losses in the actuated turbine. Further exploration of DIC under dynamic inflow conditions could also extend its applicability or potential integration with
more advanced strategies, such as wake steering control.

*Data availability.* The wind tunnel experimental data can be made available upon request.

## Appendix A: Wind speed deficit profiles

To illustrate the differences among DIC cases with increasing Strouhal number, $St \in \{0.25, 0.30, 0.40\}$, Fig. A1 presents horizontal wake profiles of WT1. Each row depicts the wind speed deficit at hub height $(1 - u/u_\infty)$ across different inflow
conditions at downstream locations $x/D \in \{2, 4, 5\}$. Similarly, Fig. A2 displays wind speed deficit profiles of WT2, comparing cases with and without WT1 actuation at downstream locations $x'/D \in \{2, 4\}$.

## Appendix B: Power detriment in a full-scale actuated turbine

To address the higher power losses experienced by the model wind turbine due to DIC, aeroelastic simulations were performed in OpenFAST v3.4.1 (Jonkman et al., 2023) using the NREL–5 MW reference turbine ($D = 126$ m). All simulations were
conducted at $u_\infty = 7.0 \, \text{m s}^{-1}$ to match the wind tunnel conditions. TurbSIM (Jonkman, 2009) was used to generate ABL inflows with the same power-law exponent and turbulence intensity values described in Sect. 2.5. Table B1 summarises the 10 min averaged power losses experienced by the actuated turbine relative to the baseline greedy case. Overall, DIC cases at $A = 2°$ exhibit power losses below $1.6\%$ across all inflow conditions, decreasing to around $0.4\%$ at $A = 1°$.





**Figure A1.** Horizontal wake profiles of WT1 showing the wind speed deficit at hub height $(1 - u/u_\infty)$ under greedy and DIC cases. Each row corresponds to a different inflow type at downstream distances $x/D \in \{2, 4, 5\}$.

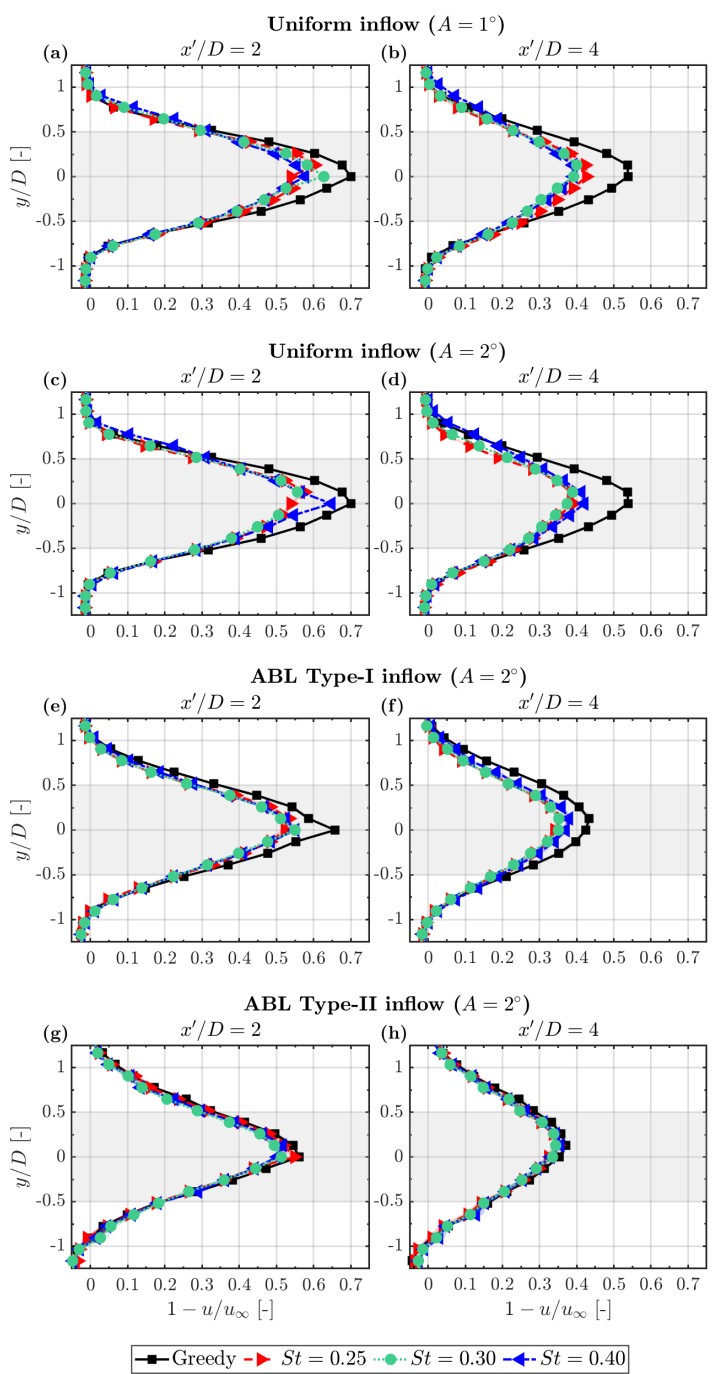

**Figure A2.** Horizontal wake profiles of WT2 showing the wind speed deficit at hub height $(1 - u/u_\infty)$. In all cases, WT2 remains in greedy operation, while WT1 alternates between greedy and DIC modes. Each row corresponds to a different inflow type at downstream distances $x'/D \in \{2, 4\}$.



**Table B1.** Power loss at WT1 due to DIC relative to the greedy case, shown as a function of pitch amplitude and Strouhal number across different inflow conditions. Results are based on OpenFAST simulations using the NREL–5 MW reference turbine ($D = 126\,\text{m}$).

|          | Uniform inflow | | ABL Type-I | ABL Type-II |
| -------- | ------------------ | ------------------ | ------------------ | ------------------ |
| $St$ [-] | $A = 1°$ | $A = 2°$ | $A = 2°$ | $A = 2°$ |
| 0.25     | -0.37 | -1.49 | -1.47 | -1.34 |
| 0.30     | -0.39 | -1.51 | -1.49 | -1.37 |
| 0.40     | -0.41 | -1.55 | -1.53 | -1.39 |

*Author contributions.* MAZI designed and conducted the measurement campaign, analysed the data and wrote the paper. PH assisted during the measurement campaign and data analysis. VP helped to design the experiments, analyse the data and structure the paper. MK supervised the research. All co-authors contributed with valuable discussions and reviewed the manuscript.

*Competing interests.* The authors declare that they have no conflict of interest.

*Acknowledgements.* This research is partly funded by the German Academic Exchange Service (DAAD) and by the Federal Ministry for Economic Affairs and Energy according to a resolution by the German Federal Parliament in the scope of research project DFWind (Ref. Nr. 0325936C). We thankfully acknowledge Anantha Padmanabhan Kidambi Sekar, Apostolos Langidis, David Onnen, Juan Manuel Boullosa Novo, Julian Jüchter, Lars Neuhaus, Marijn Floris van Dooren and Raghawendra Joshi for their assistance during the measurement campaign.





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
