# Peer review of "Dynamic induction control for mitigation of wake-induced power losses: a wind tunnel study under different inflow conditions"

_Wind Energy Science, 2024_

## Author Comment (AC1)

Dear Jennifer King and Paul Fleming,

We are pleased to submit for consideration our revised manuscript "Dynamic induction control for mitigation of wake-induced power losses: a wind tunnel study under different inflow conditions" (wes-2024-171).

We would like to sincerely thank the editorial team and the referees for their constructive feedback with valuable suggestions, which have significantly contributed to enhancing the quality and clarity of our paper.

All comments raised by the referees have been individually addressed in the detailed response below. The referees' comments are listed in order of appearance in black font, followed by our corresponding **answers** or clarifications in green font, along with **actions** taken in the manuscript in blue font. Line, figure and table numbers in the referees' comments are according to the initial manuscript. Language, spelling, style and other content refinements have been incorporated directly into the revised manuscript. All updated figures are included in the authors' response.

We believe that the revised manuscript now meets the standards required for publication in Wind Energy Science journal. Please feel free to contact us if there are any remaining questions or if further clarification is needed.

Sincerely,

On behalf of all the authors,

Manuel Zúñiga

PhD Student

**Referee 1**

This paper describes an experimental study on dynamic induction control (DIC) to mitigate wind farm wake losses using physical models in a wind tunnel. The effect is found to be mostly dependent on inflow turbulence and the amplitude of the thrust oscillation. This study provides motivation for exploring DIC further for its application at full scale wind farms.

We sincerely thank the referee for their time and effort in thoroughly reviewing our manuscript. Below, we provide detailed responses to each comment and explain how we have addressed them in the revised manuscript.

Technical comments:

1. Please archive all code and data used to generate the results in a repository like Figshare or Zenodo and cite in the paper for the sake of reproducibility. This archive does not need to be cleaned up and fully-documented/user-friendly, but it should contain all files used in the study. Please also include the OpenFAST input/output files and plotting scripts used for the results shown in the appendix.

An online database is currently being prepared including most relevant datasets and the corresponding DOI will be provided in the final publication.

2. Line 171: Is "Prandtl tube" intended to be "Pitot tube?"

A Pitot tube only measures total (stagnation) pressure, whereas a Prandtl tube (also called Pitot-static tube) measures both total and static pressure. In our experimental setup, we used a Prandtl tube to determine the inflow wind speed from the dynamic pressure (i.e. pressure difference) using Bernoulli's principle. Hence, we have adhered to the term "Prandtl tube".

3. Is it possible to estimate pitch motor energy consumption to factor that into the overall energy gain?

This could be an interesting analysis. However, we anticipate that the power gains achieved by downstream turbines would significantly outweigh the energy consumption associated with blade-pitch actuation, as DIC is designed for very low-frequency actuation within the partial load region. Moreover, this analysis falls outside the scope of our study, as the pitching mechanism of our model wind turbine is not scaled to account for factors such as pitch motor consumption, pitch inertia, and other potential losses. Such an assessment would require either numerical simulations or experiments with a full-scale turbine.

4. Line 194: Can you explain why the dataset became faulty?

The raw WindScanner files were corrupted likely due to human error when loading the scanning trajectory for the corresponding downstream distance (x/D = 9). Unfortunately, this mistake was discovered after the wind tunnel campaign was over, and the data could not be postprocessed. Nevertheless, the uniformity of the inflow condition throughout the measurement domain has been verified (up to x/D = 16) in previous wind tunnel experiments (Hulsman et al., 2022b). Hence, it can be assumed that the uniform inflow condition in our experiments at x/D = 9 remained stable.

5. **Figure 5:** It would be valuable to know some indication of the uncertainty of these statistics.

To further elucidate the uncertainty in the measured wind fields, we have added a dedicated section to the appendix, as follows:

Appendix A: WindScanner measurement uncertainty

To illustrate the uncertainty in wake measurements obtained with the lidar WindScanner, Fig. A1a-c presents contours of the relative error  $(e_u/u)$  introduced by the single-Doppler reconstruction of u at downstream distances  $x/D \in \{2, 5, 9\}$ , while Fig. A1d-f shows the corresponding statistical uncertainty at the same locations. The displayed contours correspond to experiments under uniform inflow with WT1 actuation at St = 0.30 and A =2°. The analysis of  $e_u/u$  follows the standard uncertainty propagation method (JCGM, 2008; van Dooren et al., 2017; Hulsman et al., 2022b), expressed as:

$$e_{u} = \sqrt{\left(\frac{\partial u}{\partial v_{\text{los}}}\delta_{v_{\text{los}}}\right)^{2} + \left(\frac{\partial u}{\partial v}\delta_{v}\right)^{2} + \left(\frac{\partial u}{\partial w}\delta_{w}\right)^{2} + \left(\frac{\partial u}{\partial \phi}\delta_{\phi}\right)^{2} + \left(\frac{\partial u}{\partial \theta}\delta_{\theta}\right)^{2}}$$

where  $\delta_{v_{\text{los}}}$  is the uncertainty of the measured line-of-sight wind speed, assumed to be 1% of  $u_{\infty}$ .  $\delta_{v}$  and  $\delta_{w}$  represent the uncertainties arising from neglecting the v and wcomponents, conservatively assumed as 1 ms-1.  $\delta_{\phi}$  and  $\delta_{\theta}$  refer to the uncertainties in the azimuth and elevation angles, each assumed to be 0.5 mrad. Additionally, the statistical uncertainty is expressed in terms of the margin of error  $e_{\text{MOE}} = z_{\gamma} \sigma / \sqrt{N}$ , where  $z_{\gamma} = 1.96$  is the quantile corresponding to a 95 % confidence interval,  $\sigma$  is the standard deviation of the measurements and N is the sample size.

In general,  $e_u/u$  is higher within the wake and closer to the rotor, since the wind speed deficit is more pronounced. Similarly, a higher  $e_{MOE}$  is observed in regions of higher turbulence due to increased fluctuations within the wake.

Figure A1. Uncertainty in the streamwise velocity component measured with the WindScanner at downstream distances x/D  $\in$  {2, 5, 9}. The top row (a-c) shows the relative error ( $e_u/u$ ) introduced by the single-Doppler reconstruction, while the bottom row (d-f) presents the statistical margin of error ( $e_{MOE}$ ). All wake contours correspond experiments under uniform inflow with upstream turbine actuation at St = 0.30 and A = 2°.

6. What is the mechanism by which wake recovery is enhanced with DIC, i.e., is there a difference in the mean flow structure or is this all turbulent transport? The paper mentions vortex rings, which are certainly different from isotropic homogeneous turbulence. Do these vortex rings have any significance, or is it simply a means to inject energy that breaks down into something like isotropic turbulence to increase transport?

To give further insights, we have extended the analysis of turbulence development in the wake of WT1 in Sect. 3.1.2, focusing on the downstream evolution of local power spectra at the wake centre. Specifically, the following text has been added:

As a second metric, Fig. 8 compares the downstream evolution of the local power spectra  $\phi_{\mu'}$  along the wake centreline y/D = 0 for the greedy case and two DIC cases with the same St = 0.30 but different amplitude. This provides insight into the wake energy distribution across different turbulence scales, as well as the presence of coherent structures. The power spectra are computed with the standard Welch's algorithm in MATLAB (2024), using the local time series of the wind speed fluctuations,  $u'(y) = u(y) - \langle u(y) \rangle$ . The frequency axis is expressed in terms of the dimensionless Strouhal number. In general, the energy content remains largely unchanged across all downstream locations for the baseline greedy case, consistent with the slow turbulence build up observed in the local TI profiles. On the other hand, the DIC cases exhibit not only higher energy spectra but also distinct peaks at the frequency of pitch actuation St = 0.30 and its higher harmonics. This indicates the presence of large-scale coherent structures, which can be associated with the emergence of vortex rings, as reported in (Munters and Meyers 2018; Yilmaz and Meyers 2018). In fact, Yılmaz and Meyers (2018) show that these structures cause an earlier wake breakdown, enhancing the momentum entrainment into the wake core. Furthermore, comparing the spectra of different amplitude cases, the convergence to a single spectrum with similar energy across all frequencies by x/D = 3 indicates an earlier transition to the far-wake region for the high amplitude case (A = 2°). The zoomed-in view of the dominant peak at St = 0.30 confirms that a turbulence plateau is reached at x/D = 3, followed by a decay at x/D = 5. In contrast, the low-amplitude case (A = 1°) exhibits a slower turbulence build-up, with the highest energy content observed at x/D = 5. Additionally, although not shown here, spectral analysis of the shear layer region at the rotor edges reveals an even more rapid turbulence development for both amplitude cases. This is reasonable since the transition to the far-wake region starts with the breakdown of the helical tip-vortex system into small-scale turbulence structures (Lignarolo et al. 2015) but also results in the formation of vortex rings due to periodic flow disturbances induced by DIC. This fuels momentum transport towards the wake core in response to a faster shear layer expansion.

Figure 8. Downstream evolution of the local power spectra of wind speed fluctuations ( $\phi_{u'}$ ) at the wake centreline (y/D = 0) of WT1 under uniform inflow conditions. For DIC cases, zoomed insets illustrate the downstream development of the dominant coherent structure at St = 0.30.

7. What is the difference between the Reynolds number of this study and a full-scale wind turbine, and how might that affect the conclusions?

The Reynolds number (*Re*) mismatch between model wind turbines and full-scale turbines is a well-known drawback when performing wind tunnel experiments. Nonetheless, several studies (e.g. Vermeer et al., 2003, Chamorro et al., 2012, Wang et al., 2021) indicate that the main wake features (e.g. wake deficit, turbulence intensity, shear stresses) can be reproduced once a minimum Re is achieved. In particular, Chamorro et al., (2012) suggest *Re* independence for rotor-based *Re* > 9.3 x104. Since all our experiments were conducted at around *Re* =  $2.9 \times 10^5$ , it is assumed that the wake generated by the model wind turbine resembles well that of full-scale turbines.

To further clarify the implication in *Re* mismatch between model wind turbines and fullscale turbines, we have added the following text to the methodology under the "Model wind turbine" subsection:

To account for scaling effects and manufacturing constraints, the blades feature an SD7003 low-Reynolds-number (*Re*) airfoil with increased chord length and tailored twist distribution along the span (Schottler et al., 2016). While this improves aerodynamic performance in the operating low-*Re* regimes typical of wind tunnel testing, the power coefficient ( $C_P$ ) remains unavoidably lower than that of full-scale turbines (Wang et al., 2021).

As well as the following text to the methodology under the "Experimental setup and measurement procedure" subsection:

All experiments were conducted in the partial load region at  $u_{\infty} = (7.0 \pm 0.1) \text{ ms}^{-1}$ , resulting in a rotor-based Reynolds number of  $Re \approx 2.9 \times 10^5$ . Although lower than that of full-scale turbines, typically  $O(10^6)$  to  $O(10^7)$ , Chamorro et al., (2012) suggest *Re* independence of the main wake statistics for rotor-based  $Re > 9.3 \times 10^4$ .

**8. Line 273: What type of low pass filter was applied?**

To clarify, we have added the following sentence:

...the raw signals are low-pass filtered at 12 Hz to remove high-frequency noise and facilitate visualisation. To eliminate phase distortion, a sixth-order Butterworth filter is applied using the filtfilt function in MATLAB (2024).

Furthermore, the values in Table 2 and Table 3 have been updated using a cut-off frequency of 250 Hz (i.e. about 12P of the once-per-revolution (1P) cyclic load). This choice filters out high-frequency noise while preserving most relevant dynamic components. Further increases in the cut-off frequency resulted in negligible variations.

We have added the following sentence:

.. The reported values represent 10-min averages, obtained after low-pass filtering the data at 250 Hz. This choice filters out high-frequency noise while preserving most relevant dynamic components. Further increases in the cut-off frequency resulted in negligible variations.

9. Line 323: The wording is confusing here. Since WT3 does not actually exist, how can it be operating in greedy mode?

This is an assumption based on the use of WT1's optimal  $C_P$  under greedy mode to calculate the virtual turbine power of WT3.

To clarify, we have reformulated the text as follows:

WT3's virtual power is estimated from the  $u_{\text{REWS}}$  computed in Sect. 3.2.1 assuming greedy operation with the same optimal efficiency as WT1 ( $C_P = 0.37$ ), without accounting for *Re* effects. Accordingly, only WT1 alternates between greedy and DIC modes, while WT2 and WT3 remain in greedy mode throughout.

10. Would it be possible to optimize tip speed ratio concurrently with pitch-actuated thrust oscillation to minimize power loss at the upstream turbine?

The baseline generator torque controller was kept active during all DIC mode experiments. This controller tries to maintain operation at the point of optimal aerodynamic efficiency ( $C_P^*$ ), which corresponds to the turbine's optimal tip speed ratio under steady-state conditions, as already detailed in section 2.3, L. 117.

However, it is worth noting that due to pitching around fine pitch angle, the average  $C_P$  value is slightly reduced, which is not accounted for in the standard  $K\omega^2$  torque control. Whether this could lead to a higher power gain is beyond the scope of this paper, as further experiments would be needed to test and validate this hypothesis.

The following text has been added to methodology under the DIC mode section:

Although pitch actuation causes fluctuations and a reduced mean in  $C_Q$ , the controller gain *K* is not updated during the experiments.

11. How are the turbines' power losses measured? Similarly, are the downstream turbine's power gains measured with REWS or from an electrical power measurement?

The power losses due to DIC activation are calculated relative to the measured power of WT1 in greedy mode for each inflow type. For WT2, power gains are determined similarly using measured power. Only WT3 is estimated based on the power of a virtual turbine, derived from the rotor-equivalent wind speed.

We have reformulated the text in Sect. 3.3 as follows:

Figure 14 displays the mean power ratio  $(P_i/P_{ref})$  for individual turbines (WT1, WT2, WT3) and the entire wind farm (WF), where  $P_i$  represents a 10-min average of the measured electrical power at WT1 and WT2, and the virtual power of WT3.  $P_{ref} = \sum_{i=1}^{3} P_{i,Greedy}$  is the total wind farm power output when all three turbines operate in greedy mode. Each subchart corresponds to a different inflow condition, with percentage values indicating the relative change for individual turbines and the whole wind farm with respect to the baseline greedy case.

Note that as suggested by Referee 2, we have also added a comparison of real-virtualvirtual turbine configuration to the appendix in the revised manuscript (cf. response to Referee 2 below), where both WT2 and WT3 are based on virtual power estimates.

**12. Line 362: Is it true that there was improved wake recovery but no improvement in waked turbine power?**

The improvement in wake recovery is demonstrated in Figs. 5, 6, 9 and 10 based on WindScanner measurements, while the increase in power is shown in Fig. 12, derived from WT2's electrical power measurements and WT3's virtual power across all DIC cases and inflow conditions. The absence of net power gains at the farm level under ABL Type-II is primarily due to the higher power losses experienced by the model wind turbine compared to those expected in full-scale turbines (cf. Appendix B in the original manuscript), as well as the reduced power gains in downstream turbines (WT2 and WT3) due to increased inflow turbulence.

For better clarity, we have added the following lines in the discussion:

...although control authority is notably reduced under ABL Type-II inflow, this study underscores the potential adaptability of DIC to realistic inflow conditions.

...the assessment of wind farm power gains in a virtual three-turbine configuration demonstrates consistent power benefits under uniform and ABL Type-I inflows. The absence of net power gains at the farm level under ABL Type-II is primarily due to the higher power losses experienced by the model wind turbine (approximately 10 %) compared to those expected in full-scale turbines. Supporting this, aerolastic simulations using the NREL–5 MW reference turbine (D = 126 m) show power losses below 1.6 % for all DIC cases under both uniform and ABL inflows (cf. Appendix D). These results are more closely aligned with reported losses from LES studies using actuator disc or actuator line models (e.g. Yılmaz and Meyers, 2018; Frederik et al., 2020a; Coquelet et al., 2022).

13. Line 407: What about main bearing loading?

The cited paper does not address the assessment of main shaft bearing loading, only pitch bearing load. Further studies are required to evaluate the impact on other load components.

14. Figure 12: Is it possible to put error bars here?

The Figure below includes error bars in grey. For the values based on the electrical power of physical turbines (WT1Real and WT2Real), the statistical margin of error was used. For the virtual turbine (WT3Virt) and the wind farm power (WF), standard error propagation was applied. Since the error bars are nearly imperceptible due long measurements with high sampling frequency, we decided to exclude them, consistent with the initial manuscript.